# Clinical Significance of Timing of Intubation in Critically Ill Patients with COVID-19: A Multi-Center Retrospective Study

**DOI:** 10.3390/jcm9092847

**Published:** 2020-09-02

**Authors:** Yong Hoon Lee, Keum-Ju Choi, Sun Ha Choi, Shin Yup Lee, Kyung Chan Kim, Eun Jin Kim, Jaehee Lee

**Affiliations:** 1Division of Pulmonary and Critical Care Medicine, Department of Internal Medicine, School of Medicine, Kyungpook National University, Daegu 41944, Korea; yhlee2020@knu.ac.kr (Y.H.L.); sunha20@knu.ac.kr (S.H.C.); shinyup@knu.ac.kr (S.Y.L.); 2Department of Internal Medicine, Daegu Veterans Hospital, Daegu 42835, Korea; tvbogo@naver.com; 3Department of Internal Medicine, Daegu Catholic University School of Medicine, Daegu 42472, Korea; solar903@chol.com

**Keywords:** COVID-19, acute respiratory distress syndrome, intubation, respiratory failure, mortality, intensive care units

## Abstract

The effect of intubation timing on the prognosis of critically ill patients with coronavirus 2019 (COVID-19) is not yet well understood. We investigated whether early intubation is associated with the survival of COVID-19 patients with acute respiratory distress syndrome (ARDS). This multicenter, retrospective, observational study was done on 47 adult COVID-19 patients with ARDS who were admitted to the intensive care unit (ICU) in Daegu, Korea between February 17 and April 23, 2020. Clinical characteristics and in-hospital mortality were compared between the early intubation and initially non-intubated groups, and between the early and late intubation groups, respectively. Of the 47 patients studied, 23 (48.9%) were intubated on the day of meeting ARDS criteria (early intubation), while 24 (51.1%) were not initially intubated. Eight patients were never intubated during the in-hospital course. Median follow-up duration was 46 days, and 21 patients (44.7%) died in the hospital. No significant difference in in-hospital mortality rate was noted between the early group and initially non-intubated groups (56.5% vs. 33.3%, *p* = 0.110). Furthermore, the risk of in-hospital death in the early intubation group was not significantly different compared to the initially non-intubated group on multivariate adjusted analysis (*p* = 0.385). Results were similar between early and late intubation in the subgroup analysis of 39 patients treated with mechanical ventilation. In conclusion, in this study of critically ill COVID-19 patients with ARDS, early intubation was not associated with improved survival. This result may help in the efficient allocation of limited medical resources, such as ventilators.

## 1. Introduction

Coronavirus disease 2019 (COVID-19), an infectious disease caused by severe acute respiratory syndrome coronavirus 2 (SARS-CoV-2), was declared a pandemic by the World Health Organization (WHO, Geneva, Switzerland) in March 2020 [1]. Since then, the global spread has continued, and as of July 2020, the cumulative number of confirmed patients has exceeded 16 million, and the death toll has reached nearly 660,000 [2]. The clinical spectrum of COVID-19 is variable, ranging from asymptomatic infection to acute respiratory distress syndrome (ARDS) and even death [3,4]. The prevalence of hypoxic respiratory failure in COVID-19 is approximately 20% [5], and recent reports of inpatients with COVID-19 showed that approximately 25% were admitted to an intensive care unit (ICU) [4,6,7]. Although it was reported that dexamethasone reduced the mortality in COVID-19 patients receiving invasive mechanical ventilation (MV) [8], due to lack of a proven effective antiviral agent to date, the timely application of MV and lung protective strategies also play an important role as a life-saving intervention in critically ill patients with COVID-19 [9]. In terms of public health, it is an important task to secure a sufficient supply of ICU beds and ventilators for potential surges in demand, especially in areas in the early phase of an outbreak [10].

In ARDS, the timing of intubation may be related to clinical outcomes. A previous study reported that ARDS patients undergoing late intubation had markedly higher mortality rates compared to those who were intubated early in the course of the illness [11]. Similarly, current treatment guidelines recommend early intubation in a controlled setting in case of worsening of respiratory status in COVID-19 patients with hypoxia [9]. However, early intubation in COVID-19 is not always beneficial. Performing unnecessary intubation in patients who otherwise would have improved without invasive MV can interfere with life-saving treatment for other, more severe patients in medical resource-limited settings [12]. In addition, since endotracheal intubation itself could be associated with an increased risk of aerosolization and transmission of the virus [13], reducing the frequency of unnecessary intubation is beneficial for healthcare worker protection. To our knowledge, no study exists on the prognostic effect of intubation timing in a subgroup of critically ill COVID-19 patients with ARDS. Therefore, we determined how the clinical characteristics and outcomes differ according to the timing of intubation in COVID-19 patients admitted to the ICU with ARDS, and we investigated whether early intubation has a survival benefit in such patients.

## 2. Materials and Methods

### 2.1. Study Design and Participants

Data were collected from consecutive hospitalized adults (≥18 years old) with laboratory-confirmed SARS-CoV-2 infection who subsequently were admitted to ICUs at the three tertiary referral hospitals in Daegu, Korea between 17 February and 23 April 2020. According to WHO guidelines [14], laboratory confirmation for SARS-CoV-2 was defined as a positive result on real-time reverse transcription-polymerase chain reaction assay of nasal and pharyngeal swabs. During the study period, all critically ill patients with COVID-19 who had ARDS during the clinical course were eligible for inclusion. Patients with a “do not intubate” order were excluded. ARDS was defined according to the Berlin definition [15]. Patients were included independent of the requirement for positive-pressure ventilation, considering that the purpose of our study was to investigate the relationship between intubation timing and prognosis, and that the natural course of ARDS does not start immediately upon intubation. Therefore, all patients with a history of acute respiratory failure within 1 week of a known clinical insult, with hypoxemia (PaO2/FiO2 ≤ 300 mmHg) and bilateral pulmonary infiltrates on chest radiograph not fully explained by heart failure or volume overload, were considered to have ARDS. The decision for ICU admission, oxygen therapy, respiratory support, and intubation was at the discretion of the attending physician. This study was approved by the institutional review board of each institution. Given the retrospective nature of our study, requirements for informed written consent were waived.

### 2.2. Data Collection and Definitions

Demographic and baseline characteristics, including age, sex, body mass index, presenting symptoms, vital signs, comorbid conditions, and initial laboratory findings, were obtained from the electronic medical records. Illness severity was evaluated using the Acute Physiological and Chronic Health Evaluation (APACHE) II, and Sequential Organ Failure Assessment scores. Septic shock was defined according to the third international consensus definitions for sepsis and septic shock (Sepsis-3) [16]. Acute cardiac injury was diagnosed if serum concentrations of cardiac troponin I (TNI) were above the upper limit of the reference range (>0.04 ng/mL). Acute kidney injury (AKI) was identified according to the definition of the Acute Kidney Injury Network [17] as an increase in serum creatinine level to ≥0.3 mg/dL, an increase in baseline serum creatinine level to ≥150%, or initiation of dialysis without a history of chronic kidney disease. Data on treatment and medical events in the ICU also were reviewed. Data on serial ventilatory parameters were not available. The number of patients who had died, been discharged, and remained admitted in the hospital as of 2 July 2020 were recorded.

### 2.3. Classification by Intubation Timing and Status

Patients were classified into two groups based on the previous study by Kangelaris et al. [11]: (1) Early intubation: intubated/mechanically ventilated and meeting ARDS criteria on the same day (within 24 h), and (2) initially non-intubated: not intubated on the day of meeting ARDS criteria. The initially non-intubated group was divided further into two subgroups: (A) never intubated: not requiring intubation throughout the entire hospital stay and (B) late intubation: not intubated on the day of ARDS diagnosis, but intubated on a subsequent study day.

### 2.4. Outcomes

The primary outcome was in-hospital mortality, and the main causes of death also were identified. Other outcome variables included ventilator-free days (VFDs), defined as the number of days alive and free of MV to hospital discharge or death, and the total number of days of ICU stay and MV application in survivors to hospital discharge.

### 2.5. Statistical Analysis

Data were expressed as medians (interquartile ranges, IQRs) for continuous variables and numbers and percentages for categorical variables. For bivariate analysis, the Mann–Whitney U test or *t*-test was used for continuous variables, and the χ2 or Fisher’s exact test for categorical variables. Survival curves were developed using the Kaplan–Meier method with log-rank test. A bivariate Cox proportional hazard model was used to adjust the effect of potential confounders on the association between intubation status and in-hospital mortality. The multivariate analysis model incorporated variables that varied according to intubation status with a *p* value < 0.05 or that were considered clinically important. Variables from laboratory tests with missing values were excluded. All statistical procedures were performed using SPSS software (version 24.0, SPSS Inc., Chicago, IL, USA) and MedCalc software (version 19.2.1, Ostend, Belgium). *p* < 0.05 was considered statistically significant when a two-tailed test was performed.

## 3. Results

### 3.1. Patient Characteristics

Of the 47 patients studied (mean follow-up 46 days; IQR, 24–86 days), 23 (48.9%) were intubated on the day of ARDS diagnosis (early intubation) and 16 (34%) were not initially intubated, but subsequently required intubation during follow-up (late intubation). The median time interval from ARDS diagnosis to intubation in the late intubation group was 3 days (IQR, 1–7 days). Eight patients (17%) were never intubated during the follow-up period. All patients in the initially nonintubated group received oxygen via high-flow nasal cannula (HFNC) either before intubation or throughout the treatment period.

Overall, median age of the 47 patients was 70 years (IQR, 63–77 years) and 28 (59.6%) were male. Demographics and baseline characteristics according to intubation status are summarized in Table 1. All patients were divided into early intubation and initially non-intubated groups for comparative analysis. In addition, a subgroup of patients treated with MV was divided into the early and late intubation groups. Age, sex, comorbid conditions, and presenting symptoms did not show significant differences between the groups. However, among the initial vital signs, respiratory rate was significantly higher in the early intubation than in the initially non-intubated groups (median, 28 breaths per minute (bpm); IQR, 22–34 vs. 21 bpm; IQR, 20–26, *p* = 0.007).

### 3.2. Laboratory Indices, Severity of Illness, and Clinical Course

Laboratory findings on hospital admission are shown in Table 2. Of all patients, creatine kinase-MB (CK-MB) was significantly higher in the early intubation than in the initially non-intubated groups (median, 1.5 U/L; IQR, 1–4.3 vs. 1.1 U/L; IQR, 0.8–1.8, *p* = 0.025). This difference also was observed between the early and late intubation groups (median, 1.5 U/L; IQR, 1.0–4.3 vs. 1.0 U/L; IQR, 0.8–1.4, *p* = 0.019). Laboratory tests other than CK-MB did not show a significant difference between the groups.

The APACHE II score was significantly higher in the early intubation than in the initially non-intubated groups (median, 15; IQR, 10–17 vs. 11; IQR, 8–14; *p* = 0.042; Table 3). On arterial blood gas testing at the time of ARDS diagnosis, the early intubation group had a significantly lower pH (median, 7.34; IQR, 7.31–7.44 vs. 7.45; IQR, 7.42–7.50; *p* = 0.001) and PaO2/FiO2 ratio (median, 86; IQR, 69–123 vs. 144; IQR, 70–206; *p* = 0.028), and higher PaCO2 (median, 37.6 mmHg; IQR, 33.3–50.2 vs. 32.2 mmHg; IQR, 26.8–36.5; *p* = 0.001) than the initially non-intubated group. In the subgroup analysis with patients treated with MV, pH and PaCO2 in the early intubation group showed significant differences compared to values in the late intubation group (median, 7.34; IQR, 7.31–7.44 vs. 7.43; IQR, 7.40–7.49; *p* = 0.013 for pH and 37.6 mmHg; IQR, 33.3–50.2 vs. 32.3 mmHg; IQR, 23.8–37.1; *p* = 0.002 for PaCO2).

Among the treatment modalities, the frequency of HFNC use was significantly lower in the early intubation compared to the initially non-intubated (56.5%; *n* = 13 vs. 100%; *n* = 24; *p* < 0.001) or late intubation groups (56.5%; *n* = 13 vs. 100%; *n* = 16; *p* = 0.002). Among the initial ventilator parameters, plateau pressure of the early intubation group was significantly lower than that of the late intubation group (median, 27 mmHg; IQR, 22–29 vs. 29 mmHg; IQR, 26–32; *p* = 0.014). During intensive treatment in the ICU, the incidence of ventilator-associated (VAP) or hospital-acquired (HAP) pneumonia was significantly higher in the early intubation than the initially non-intubated groups (30.4%; *n* = 7 vs. 4.2%; *n* = 1; *p* = 0.023). In patients treated with MV, VAP incidence tended to be higher in the early than in the late intubation groups, but there was no statistical significance (30.4%; *n* = 7 vs. 6.2%; *n* = 1; *p* = 0.109).

### 3.3. Clinical Outcomes

At the end of the study period, four patients (8.5%) remained hospitalized, 21 (44.7%) had died in the hospital, and 22 (46.8%) had been discharged. COVID-19–related ARDS was the most common cause of death (52.4%, *n* = 11), followed by VAP (19%, *n* = 4), catheter-related blood stream infection (9.5%, *n* = 2), acute myocardial infarction (9.5%, *n* = 2), and AKI (4.8%, *n* = 1). One died of unknown cause, who suffered from sudden cardiac arrest while receiving intensive care without intubation, and died after cardiopulmonary resuscitation. Among the survivors (*n* = 26), 38.5% (*n* = 10), 34.6% (*n* = 9), and 26.9% (*n* = 7) were in the early intubation, late intubation, and never intubated groups, respectively (Figure 1). Of the 19 survivors treated with MV, 94.7% (*n* = 18) were weaned from the ventilator successfully.

Data on clinical outcomes between the groups are presented in Table 4. The relevant variables were compared between the early intubation group and the other groups for all patients or patients who underwent intubation and MV. There was no statistically significant difference in in-hospital mortality between the early intubation and initially non-intubated groups (56.5% vs. 33.3%, *p* = 0.110) and between the early and late intubation groups (56.5% vs. 43.8%, *p* = 0.433). Survival curve analysis also showed that the early intubation group had no significant difference compared to the other groups (Figure 2). The findings of no significant difference in mortality rate according to the timing of intubation were observed consistently at the three institutions participating in this study (data not shown). No significant differences between the groups were noted in terms of causes of death. VFDs in the early intubation group were significantly lower than those in the initially non-intubated (median, 9 days; IQR, 0–18 vs. 28 days; IQR, 9–45; *p* = 0.008) or late intubation (median, 9 days; IQR, 0–18 vs. 25 days; IQR, 7–45; *p* = 0.033) groups. Among the survivors, there were no significant differences between the groups in terms of number of days of ICU stay or MV use.

### 3.4. Effects of Early Intubation on Mortality

A Cox proportional hazards model was used to analyze whether early intubation independently affected survival outcome. Variables included in the model were respiratory rate, arterial pH, PaCO2, PaO2/FiO2, use of HFNC, plateau pressure, VAP during ICU stay, and APACHE II score. In adjusted analysis, early intubation showed no significant effect on in-hospital mortality compared to the initially non-intubated group (adjusted hazard ratio (aHR) = 2.278, 95% confidence interval (CI) = 0.356–14.585; *p* = 0.385). The same analysis was conducted on patients treated with MV and adjusted based on variables that differed between the early and late intubation groups: respiratory rate, PaCO2, use of HFNC, plateau pressure, and APACHE II score. Likewise, early intubation had no significant survival benefit compared to late intubation (aHR = 1.964, 95% CI = 0.351–11.004; *p* = 0.442).

## 4. Discussion

Our study was conducted to examine how clinical features and outcomes differed depending on the timing of intubation and to verify whether early intubation is associated with the survival of critically ill COVID-19 patients with ARDS. Almost half of the patients had tracheal intubation on the day of meeting ARDS criteria. The disease severity of this patient subset (early intubation) tended to be higher than that of other groups. After adjusting for potential confounding variables, including severity of illness, early intubation had no survival benefit.

According to a recent systematic review of the mortality rates of patients with COVID-19 in ICUs [18], the reported overall mortality rate was 25.7%, which is lower than our findings. However, the actual mortality rate may be higher, as mortality rates from previous studies varied from 8% to 66.7% and more than half of the patients were still in the ICU at publication. Rather, considering that only 8.5% of patients were hospitalized at the time of data collection and whose definite outcome were unknown, the in-hospital mortality rate of 44.7% in our study also appeared to be a similar finding.

The main finding of our study is somewhat in conflict with the study of Kangelaris et al. [11], who showed that late intubation in critically ill non–COVID-19 patients with ARDS had a significantly higher risk of death compared to early intubation. First of all, it seems necessary to consider that age, comorbidities, and the baseline severity of illness of the study population are different. Our study population’s age was far higher than that of Kangelaris et al. (70 vs. 55 years). Additionally, the average APACHE II score in the early intubation group reported by Kangelaris et al. was 31, but in our study, the median value was 14. Such differences in study population may reflect the tendency to be admitted preemptively to the ICU for close monitoring and use of the ventilator when necessary, even if the general condition of the COVID-19 patients is slightly deteriorated because the natural clinical course has not been understood clearly to date. This suggests that our patients were intubated when in a less severe condition or earlier in the course of the disease than those of Kangelaris et al. [11]. Nevertheless, because the definition of early intubation was the same as in that study, our findings suggested that the ARDS due to COVID-19 may differ from other common causes in terms of the effect of intubation timing on prognosis.

In our study, the frequency of VAP was higher in the early intubation group than in the other groups, which may be partly related to the relatively low VFDs in the same group. Although the etiology of VAP varies, the duration of MV is known to be an important determinant for VAP development, and the risk tends to be higher, especially in the early stages of ventilation support [19]. In addition, VAP was the most common cause of death except COVID-19 itself in our study, and although there was no statistical significance, the frequency of VAP as the cause of death tended to be higher in the early intubation group than in the other groups. This supported the possibility that early intubation itself contributed to VAP risk in our patients with COVID-19, and consequently had some negative impact on clinical outcomes. On the other hand, the mortality rate of the late intubation group did not show a significant difference compared to that in the early intubation group, and all of the never-intubated patients survived using HFNC except one case of sudden death. Our findings can help to allocate and reserve ventilators more efficiently in clinical settings where COVID-19 confirmed cases are rapidly increasing.

Beneficial effects of HFNC in critically ill patients with hypoxemic respiratory failure have been suggested in several studies [20,21,22]. A recent report also showed the potential for HFNC to be successful as a first-line treatment in ARDS [23]. In our study, more than half of the survivors initially were non-intubated, and a quarter survived without endotracheal intubation, all of whom used HFNC as the initial oxygen supply. These results suggested that HFNC also can be useful in COVID-19 patients with ARDS. In addition, one report showed that the risk of air or contact surface contamination by HFNC was not higher compared to a conventional oxygen mask, and patients with HFNC tend to be more comfortable [24,25]. Therefore, in terms of managing an ICU that cares for COVID-19 patients, securing a sufficient number of HFNCs seems as important as the procurement of ventilators.

Because of the need for adherence to airborne precautions and personal protective equipment [26], medical staff involved in the management of patients with COVID-19 find cases difficult to deal with quickly in the event of a sudden deterioration. Moreover, emergency intubation may increase the risk of nosocomial infection of healthcare providers, so treatment guidelines recommend early intubation in a controlled setting if respiratory status worsens [9]. Therefore, our results should be applied carefully in clinical practice, and a predictive model that can identify critically ill COVID-29 patients at risk for respiratory deterioration that requires intubation is needed. Among the patients who initially were non-intubated in our study, the initial vital signs and severity scoring systems, which were readily available, were not significantly different between the never-intubated and late intubation groups (Appendix A). Although there were differences in some laboratory tests, it seems difficult to attach meaning due to the small sample size. Thus, further research with a large number of patients on this subject is necessary to properly screen for patients requiring proactive intubation

Our study has several limitations. First, this is a retrospective study that included only 47 critically ill patients. Patients in the early intubation group had higher disease severity, which was estimated to have had a significant effect on the relatively higher mortality rate of those patients, and the difference in mortality rates seems not to be statistically significant due to the small sample size. Although we attempted to adjust variables including the severity of illness by multivariate analysis, there may be a statistical limitation related to sample size, which should be taken into account when interpreting our findings. Second, no serial ventilator data were available. If there were intergroup differences in adherence to lung protective ventilation strategies, which could affect treatment outcomes, it could have influenced our findings as a confounder. Third, some laboratory tests had missing values and were excluded from multivariate analysis. Fourth, data on long-term outcomes, such as pulmonary function or quality of life after discharge, were not available. Despite these limitations, our findings can be an important reference for COVID-19 critical care, if validated in future large-scale studies.

In summary, in this study of critically ill patients with COVID-19 and ARDS, more than half of the survivors were not intubated on the day of meeting ARDS criteria, and some were never intubated. There were no significant differences in in-hospital mortality between the early intubation group and the other groups. Furthermore, after adjustment for possible confounding factors, early intubation was not associated with improved survival. Our results may help in the efficient allocation of limited medical resources, such as ventilators.

## Figures and Tables

**Figure 1 jcm-09-02847-f001:**
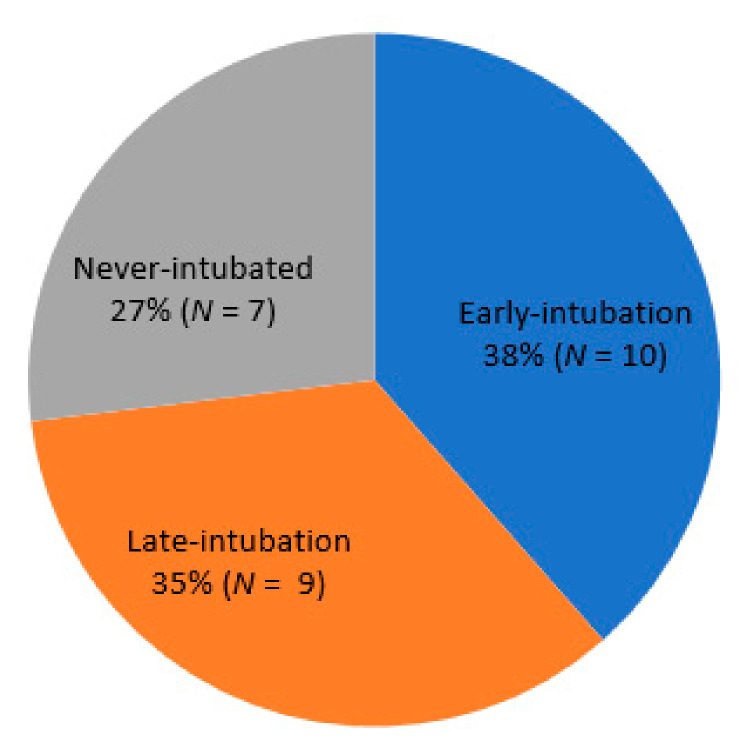
Proportion of intubations implemented among 26 patients who survived after intensive care for COVID-19 with ARDS.

**Figure 2 jcm-09-02847-f002:**
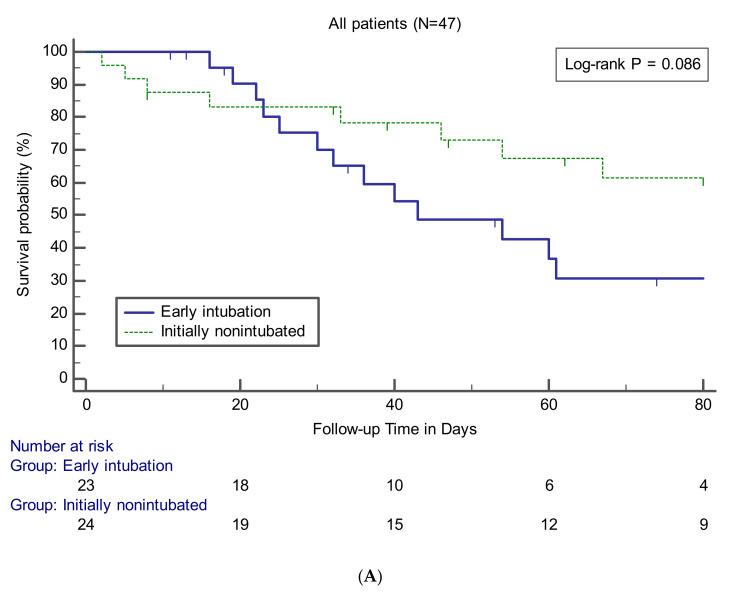
Kaplan–Meier curves showing survival probability during follow-up in patients (**A**) or in patients treated with MV (**B**).

**Table 1 jcm-09-02847-t001:** Demographics and baseline characteristics of critically ill COVID–19 patients with ARDS.

Variables	Early Intubation (*n* = 23)	Initially Nonintubated (*n* = 24)	*p* Value ^a^	Late Intubation (*n* = 16)	*p* Value ^b^
Age	72 (64–76)	69 (60–78)	0.655	66 (59–77)	0.475
Male	14 (60.9)	14 (58.3)	0.859	10 (62.5)	0.918
Body mass index, kg/m^2^	22.8 (21.0–26.7)	25.6 (22.5–27.1)	0.167	25.1 (22.7–27.3)	0.241
Comorbidities					
Any comorbidities	16 (69.6)	19 (79.2)	0.450	13 (81.2)	0.480
Hypertension	10 (43.5)	11 (45.8)	0.871	8 (50)	0.688
Diabetes	10 (43.5)	8 (33.3)	0.474	7 (43.8)	0.987
Chronic kidney disease	1 (4.3)	2 (8.3)	>0.999	2 (12.5)	0.557
Dementia	2 (8.7)	3 (12.5)	>0.999	1 (6.2)	>0.999
Cerebrovascular disease	0 (0)	2 (8.3)	0.489	1 (6.2)	0.410
Malignancy	2 (8.7)	4 (16.7)	0.666	4 (25)	0.205
Cardiovascular disease	4 (17.4)	4 (16.7)	>0.999	2 (12.5)	>0.999
Chronic lung disease	3 (13.0)	1 (4.2)	0.348	1 (6.2)	0.631
Chronic liver disease	1 (4.3)	2 (8.3)	>0.999	0 (0)	>0.999
Duration of symptoms before admission, days	7 (5–11)	5 (4–12)	0.280	5 (3–10)	0.143
Presenting symptoms					
Fever	18 (78.3)	16 (66.7)	0.374	9 (56.2)	0.174
Dyspnea	19 (82.6)	17 (70.8)	0.341	12 (75)	0.694
Cough	12 (52.2)	14 (58.3)	0.671	8 (50)	0.894
Sputum	10 (43.5)	10 (41.7)	0.900	6 (37.5)	0.709
Myalgia	5 (21.7)	5 (20.8)	>0.999	4 (25)	>0.999
Fatigue	3 (13.0)	7 (29.2)	0.286	6 (37.5)	0.123
Diarrhea	2 (8.7)	5 (20.8)	0.416	2 (12.5)	>0.999
Vital signs at the time of ICU admission					
Mean arterial pressure, mmHg	93 (90–107)	93 (86–102)	0.539	93 (86–97)	0.388
Heart rate, beats/min	85 (76–124)	88 (80–100)	0.915	92 (74–100)	0.808
Respiratory rate, breaths/min	28 (22–34)	21 (20–26)	0.007	21 (20–29)	0.057
Body temperature, ℃	37.3 (36.4–37.8)	36.8 (36.5–37.4)	0.781	36.9 (36.6–37.4)	0.863

Data are presented as median (interquartile range) or *n* (%). ^a^ Comparison between the early intubation and initially non-intubated groups. ^b^ Comparison between the early and late intubation groups. Abbreviation: ARDS, acute respiratory distress syndrome; ICU, intensive care unit.

**Table 2 jcm-09-02847-t002:** Initial laboratory findings of critically ill COVID–19 patients with ARDS.

Variables	Early Intubation (*n* = 23)	Initially Nonintubated (*n* = 24)	*p* Value ^a^	Late Intubation (*n* = 16)	*p* Value ^b^
White blood cells, 10^3^/L	6.89 (4.73–9.82)	7.16 (5.21–9.55)	0.907	7.16 (5.9–9.96)	0.679
Hemoglobin, g/dL	13.0 (11.2–14.4)	13.2 (11.5–14.4)	0.935	13.1 (10.9–14.1)	0.828
Hematocrit, %	37.6 (33.1–42)	39.3 (33.2–41.2)	0.849	38.7 (31.8–40.6)	0.706
Platelets, 10^3^/L	197 (146–296)	211 (156–295)	0.702	220 (156–295)	0.396
C–reactive protein, mg/dL	9.98 (5.66–15.59)	10.27 (6.46–13.83)	0.983	11 (7.39–17.43)	0.668
Procalcitonin, mmol/L	0.27 (0.12–0.74)	0.11 (0.1–0.21)	0.044	0.13 (0.1–0.21)	0.082
Lactate, mmol/L	1.8 (1.3–2.5)	1.6 (1.3–2.3)	0.557	1.8 (1.3–2.4)	0.947
Albumin, g/dL	3.3 (3–3.5)	3.4 (3.3–3.5)	0.298	3.4 (3.3–3.7)	0.329
AST, U/L	48 (35–81)	50 (35–91)	0.717	50 (33–65)	0.875
ALT, U/L	27 (19–34)	20 (13–57)	0.302	22 (13–53)	0.346
Total bilirubin, mg/dL	0.62 (0.4–0.85)	0.54 (0.32–0.83)	0.537	0.6 (0.3–0.83)	0.607
BUN, mg/dL	15.7 (12.5–24.7)	15.6 (8.9–22.6)	0.609	16.8 (11.8–35.3)	0.842
Creatinine, mg/dL	0.9 (0.7–1.2)	0.9 (0.67–1.45)	0.741	0.95 (0.7–1.9)	0.484
Sodium, mmol/L	134 (132–139)	137 (133–138)	0.407	137 (132–139)	0.330
Potassium, mmol/L	3.9 (3.3–4.3)	4 (3.3–4.7)	0.327	3.9 (3.3–4.6)	0.427
Glucose, mg/dL	156 (117–197)	139 (111–168)	0.312	161 (122–172)	0.966
LDH, U/L	486 (410–559)	442 (370–632)	0.606	468 (344–698)	0.818
D-dimer, ug/mL	2.0 (0.77–3.87)	2.5 (1.23–6.4)	0.447	2.03 (1.07–3.74)	0.642
Prothrombin time, INR	1.13 (1.02–1.27)	1.08 (1.02–1.26)	0.910	1.08 (1.02–1.27)	0.908
NT–proBNP, pg/mL	826 (243–1376)	570 (316–1395)	0.648	540 (372–2026)	0.917
Troponin I, ng/mL	0.03 (0.02–0.18)	0.02 (0.01–0.02)	0.073	0.02 (0.01–0.02)	0.114
CK–MB, U/L	1.5 (1–4.3)	1.1 (0.8–1.8)	0.025	1 (0.8–1.4)	0.019

Data are presented as median (interquartile range). ^a^ Comparison between the early intubation and initially non-intubated groups. ^b^ Comparison between the early and late intubation groups. Abbreviation: ARDS, acute respiratory distress syndrome; AST, aspartate aminotransferase; ALT, alanine aminotransferase; BUN, blood urea nitrogen; LDH, lactate dehydrogenase; NT–proBNP, *n*–terminal probrain natriuretic peptide; CK–MB, creatine kinase–MB.

**Table 3 jcm-09-02847-t003:** Severity of illness and clinical course of critically ill COVID–19 patients with ARDS.

Variables	Early Intubation (*n* = 23)	Initially Nonintubated (*n* = 24)	*p* Value ^a^	Late Intubation (*n* = 16)	*p* Value ^b^
Severity of illness on ICU admission					
Septic shock	4 (17.4)	2 (8.3)	0.416	1 (6.2)	0.631
Acute kidney injury	7 (30.4)	5 (20.8)	0.450	4 (25)	>0.999
Acute cardiac injury	8 (34.8)	3 (12.5)	0.071	2 (12.5)	0.152
SOFA score	3 (2–7)	2 (2–4)	0.134	3 (2–4)	0.336
APACHE II score	15 (10–17)	11 (8–14)	0.042	14 (8–15)	0.252
ABGA at the time of diagnosis of ARDS					
pH	7.34 (7.31–7.44)	7.45 (7.42–7.5)	0.001	7.43 (7.4–7.49)	0.013
PaCO_2_, mmHg	37.6 (33.3–50.2)	32.2 (26.8–36.5)	0.001	32.3 (23.8–37.1)	0.002
PaO_2_, mmHg	77.3 (55.3–85)	67.8 (55–82.3)	0.389	67.8 (55.7–79.7)	0.339
HCO_3_, mmol/L	22.6 (21.1–25.4)	22.4 (18.8–25.5)	0.672	20.8 (17.4–26.3)	0.259
PF ratio	86 (69–123)	144 (70–206)	0.028	120 (62–188)	0.204
ICU management					
HFNC	13 (56.5)	24 (100)	<0.001	16 (100)	0.002
NM blockade	15 (65.2)	9 (37.5)	0.057	9 (56.2)	0.571
CRRT	5 (21.7)	5 (20.8)	>0.999	5 (31.2)	0.711
Tracheostomy	9 (39.1)	7 (29.2)	0.471	7 (43.8)	0.773
ECMO	3 (13.0)	4 (16.7)	>0.999	4 (25)	0.415
Medical treatment					
Antiviral agents					
Lopinavir-ritonavir	20 (87.0)	16 (66.7)	0.101	11 (68.8)	0.235
Darunavir–cobicistat	3 (13.0)	7 (29.2)	0.286	5 (31.2)	0.235
Antibiotics	23 (100)	24 (100)		16 (100)	
Hydroxychloroquine	20 (87.0)	22 (91.7)	0.666	14 (87.5)	>0.999
Glucocorticoid	18 (78.3)	19 (79.2)	>0.999	15 (93.8)	0.370
Medical event during ICU care					
Septic shock	20 (87.0)	15 (62.5)	0.055	14 (87.5)	>0.999
Acute kidney injury	10 (43.5)	7 (29.2)	0.307	7 (43.8)	0.987
Acute cardiac injury	10 (43.5)	5 (20.8)	0.096	4 (25)	0.237
VAP or HAP	7 (30.4)	1 (4.2)	0.023	1 (6.2)	0.109
CRBSI	4 (17.4)	3 (12.5)	0.701	3 (18.8)	>0.999
Bleeding	3 (13.0)	3 (12.5)	>0.999	3 (18.8)	0.674
CPCR	1 (4.3)	3 (12.5)	0.609	2 (12.5)	0.557

Data are presented as median (interquartile range) or *n* (%). ^a^ Comparison between the early intubation and initially non-intubated groups. ^b^ Comparison between the early and late intubation groups. Abbreviation: ARDS, acute respiratory distress syndrome; ICU, intensive care unit; SOFA, Sepsis–related Organ Failure Assessment; APACHE II, Acute Physiology and Chronic Health Evaluation; ABGA, arterial blood gas analysis; PF ratio, arterial partial pressure of oxygen (PaO2)/fraction of inspired oxygen (FiO2) ratio; MV, mechanical ventilation; PEEP, positive end expiratory pressure; HFNC, high–flow nasal cannula; NM blockade, neuromuscular blockade; CRRT, continuous renal replacement therapy; ECMO, extracorporeal membrane oxygenation; VAP, ventilator–associated pneumonia; HAP, hospital–acquired pneumonia; CRBSI, catheter–related bloodstream infection; CPCR, cardiopulmonary–cerebral resuscitation.

**Table 4 jcm-09-02847-t004:** Clinical outcomes of critically ill COVID–19 patients with ARDS.

Variables	Early Intubation (*n* = 23)	Initially Nonintubated (*n* = 24)	*p* Value ^a^	Late Intubation (*n* = 16)	*p* Value ^b^
In-hospital mortality	13 (56.5)	8 (33.3)	0.110	7 (43.8)	0.433
Main cause of death					
COVID-19 related ARDS	7/13 (53.8)	4/8 (50)	>0.999	4/7 (57.1)	>0.999
VAP	3/13 (23.1)	1/8 (12.5)	>0.999	1/7 (14.3)	>0.999
CRBSI	2/13 (15.4)	0/8 (0)	0.505	0/7 (0)	0.521
Acute kidney injury	0/13 (0)	1/8 (12.5)	0.381	1/7 (14.3)	0.350
Myocardial infarction	1/13 (7.7)	1/8 (12.5)	0.999	1/7 (14.3)	>0.999
Unknown ^c^	0/13 (0)	1/8 (12.5)	0.381	0/12 (0)	
Ventilator-free days	9 (0–18)	28 (9–45)	0.008	25 (7–45)	0.033
ICU days ^d^	13 (7–33)	14 (7–48)	0.691	47 (13–74)	0.101
Days of MV ^d^	10 (4–24)	6 (0–25)	0.325	20 (9–57)	0.177

Data are presented as median (interquartile range) or *n* (%). Median duration of follow–up was 46 days (IQR, 24–86 days). Abbreviation: ARDS, acute respiratory distress syndrome; VAP, ventilator–associated pneumonia; CRBSI, catheter–related blood stream infection; ICU, intensive care unit; MV, mechanical ventilation. ^a^ Comparison between the early intubation and initially non-intubated groups. ^b^ Comparison between the early and late intubation groups. ^c^ A patient in the never-intubated group died suddenly due to unknown cause. ^d^ Among survivors to hospital discharge.

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
