# Peer review of "Clinical Significance of Timing of Intubation in Critically Ill Patients with COVID-19: A Multi-Center Retrospective Study"

_jcm, 2020, doi:10.3390/jcm9092847_

Round 1
Reviewer 1 Report
Dear Editor,
The study “Does early intubation improve the survival of critically ill patients with COVID-19?” touches on an important topic about the timing of intubation in COVID patients with acute respiratory failure. Authors had not found any clinical benefits in the group intubated immediately after admission comparing to the group with delayed decision about starting ventilation. Moreover, a substantial number of patients avoided intubation at all. The results of the study suggest that early intubation should not be a routine approach for every patients and more individualized approach is more reasonable. In spite of the retrospective design of the study I have found the study interesting and worthy of note, especially given the paucity of data in this important field.
The study is clinically relevant, methodologically is correct and properly performed. The cohort is very well characterized. I cannot see any important methodological drawbacks
I can only report a few points of doubt or uncertainty, which may be addressed by authors:
- Table 1 – I do not understand why in the second column of “early intubation” there are only 22 subjects.
- The name of the subgroups of late intubation group is not defined in the Methods section. Moreover, it can be misleading to give the same name (“early intubation”) to the group of patients intubated at the first day and the group of patients intubated later on. I would propose to rename and define these subgroups.
- Although mortality is the most important outcome, it would be interesting to assess other outcomes like: the risk of SARS-CoV-2 transition to staff or costs of treatment. These outcome are also important factors for making clinical decisions. Is it possible to add these data: rough estimation of the cost of treatment in both groups and were there any incidences of COVID among staff ?
- I could not find in Methods the rational for early intubation or HFNC therapy in the patients admitted to hospitals. Weather, the decision was left to the discretion of a doctor on duty or there was some kind of hospital recommendations. It should be elaborated.
- In the “never ventilated” group one patient died without being intubated and ventilated. It should be clearly stated in results, not only mentioned in discussion.
Author Response
Point 1: Table 1 – I do not understand why in the second column of “early intubation” there are only 22 subjects.
Response 1: We apologize for this mistake. This error happened during the process of organizing the contents of the table before submission. We meticulously rechecked all the data and revised throughout the entire manuscript.
At this point, we think that two columns for early intubation group seem to be redundant. Thus, we have revised all the tables with repetitive early intubation group column omitted.
Point 2: The name of the subgroups of late intubation group is not defined in the Methods section. Moreover, it can be misleading to give the same name (“early intubation”) to the group of patients intubated at the first day and the group of patients intubated later on. I would propose to rename and define these subgroups.
Response 2: We are sorry to make you misunderstood. First, there are two main groups (early intubation vs. initially nonintubated groups) in this study. Additionally, there are two subgroups (late intubation vs. never intubated groups) for the initially nonintubated group.
In the “2.3. Classification by intubation timing and status” section, definitions of each group are described.
Point 3: Although mortality is the most important outcome, it would be interesting to assess other outcomes like: the risk of SARS-CoV-2 transition to staff or costs of treatment. These outcome are also important factors for making clinical decisions. Is it possible to add these data: rough estimation of the cost of treatment in both groups and were there any incidences of COVID among staff ?
Response 3: Thank you for your valuable comment. There were no cases of SARS-CoV-2 transition to medical staff in all three institutions. And, unfortunately, we could not obtain data on the cost of treatment for study patients.
Point 4: I could not find in Methods the rational for early intubation or HFNC therapy in the patients admitted to hospitals. Weather, the decision was left to the discretion of a doctor on duty or there was some kind of hospital recommendations. It should be elaborated.
Response 4: Thank you for your suggestion. Every clinical decision was at the discretion of the attending physician. We have added a sentence to the “2.1. study design and participants” section as follows:
“The decision of ICU admission, oxygen therapy, respiratory support, and intubation was at the discretion of the attending physician” [page 2, lines 87-88]
Point 5: In the “never ventilated” group one patient died without being intubated and ventilated. It should be clearly stated in results, not only mentioned in discussion.
Response 5: Thank you for your suggestion. We have added a description of a deceased patient in the never intubated group to the Results section as follows:
“One died of unknown cause, who suffered from sudden cardiac arrest while receiving intensive care without intubation, and died after cardiopulmonary resuscitation.” [page 8, lines 204-206]

Reviewer 2 Report
This study by Lee, et al seeks to determine the “effect” of early vs late intubation on mortality in critically ill patients with respiratory failure from COVID. They retrospectively analyzed 47 patients admitted to the ICU with ARDS who were either intubated on the day of admission (“early”), intubated after the first day of admission (“late”) or never intubated. The early intubation group was clearly sicker at the time of admission by almost all measured parameters and ultimately had the higher mortality (56.5% vs 33.3%) although these results were not statistically significant (but this seems to be more related to small sample size than anything else).
The authors conclude that Unfortunately, I do not believe that the conclusions that the authors draw here are valid based on the data presented. In this retrospective observational study, the authors can only speak to the clinical outcomes related to early vs late intubation, not the effect of intubation timing on mortality. Based on the data presented, the early intubation group and late intubation group were COMPLETELY different populations in whom the decision to intubate was made when clinically appropriate. There is no link that I can see between the timing of intubation and mortality and can certainly not be said that early intubation causes harm.
In the discussion, the authors state that “even if a critically ill COVID-19 patient exhibits acute hypoxemia that meets ARDS criteria, close observation without intubation may not be inferior to MV after preemptive intubation.” This type of conclusion is not supported by this data and can only be addressed in a prospective manner.
Abstract
Line 18 – cannot say “affects survival” in a retrospective study, can only say “is associated with survival”
Line 22 – you do not define “non-early” intubation, only “late intubation” in the manuscript. Need to be consistent.
Line 27 – should report the actually mortality rates in the early group and initially non-intubated group. Not just the p-value.
Introduction
Line 46 – consider updating this to reflect the effectiveness of dexamethasone in critically ill patients with COVID.
Line 52 – you cannot quote the Kangelaris study as reporting that “delayed intubation had markedly higher mortality rates.” Never does that study suggest that the late intubation group was synonymous with delayed intubation. Delayed intubation suggests that intubation should have been performed early but was not.
Materials and Methods
Line 99 – was “same day” within a 24 hour period, starting at time of study inclusion? Or was this within 24 hours of hospital admission, ICU admission or other?
Table 1
I am confused about some of the reported p-values in this table. Take the diabetes and chronic kidney disease rows, for example. For diabetes, there are 10 patients in the early group and 8 in the initially nonintubated group with a p-value of 0.47. For kidney disease, there are 10 patients in the early group and only 2 in the initially nonintubated group which makes for a greater difference, but a much larger p-value. Am I misinterpreting the data here? Should the p-value be smaller for this 43.5% vs 8.3% difference?
Results
Line 179 Clinical outcomes – can you report the time to intubation in the late intubation group?
Figure 1 – can you explain what point you want to get across with this Figure?
Line 193-194 – be clear about which groups you are referring to in the parentheses (early vs initially nonintubated and early vs late intubation).
Line 198 – should read “in” terms of
Figure 2 – in black and white, the legend is uninterpretable.
Line 216 – Because the higher mortality in the early group is fully explained by their higher severity of illness at presentation, I am not sure of the value of this analysis.
Discussion
Line 228 – This retrospective study design cannot determine whether timing of intubation “affects” survival.
Line 265-267 – I think that this statement goes too far and is not supported by data.
Line 275-276 – In the initially nonintubated group, if both the never intubated and the late intubated groups were both on HFNC, can you say that HFNC “may help avoid unnecessary intubation?”
297-299 – You repeatedly state that “no significant difference in mortality between the early intubation and the other groups was observed.” However, to me there was a clear trend towards higher mortality in the early intubation group than the late intubation group (56 vs 33%) that was driven by higher disease severity and only non-significant because of small sample size. This should probably be mentioned.
Author Response
Abstract
Point 1: Line 18 – cannot say “affects survival” in a retrospective study, can only say “is associated with survival”
Response 1: Thank you for helpful comment. In accordance with your suggestion, we have corrected the sentence as follows:
“We investigated whether early intubation is associated with survival of COVID-19 patients with acute respiratory distress syndrome (ARDS)” [page 1, lines 20-21]
Point 2: Line 22 – you do not define “non-early” intubation, only “late intubation” in the manuscript. Need to be consistent.
Response 2: We agree with your opinion. We have deleted the word “non-early” and corrected the sentence as follows:
“Clinical characteristics and in-hospital mortality were compared between the early intubation and initially nonintubated groups, and between the early and late intubation groups, respectively.” [page 1, lines 25-26]
Point 3: Line 27 – should report the actually mortality rates in the early group and initially non-intubated group. Not just the p-value.
Response 3: We have added mortality rate as you pointed out.
“No significant difference in in-hospital mortality rate was noted between the early group intubation and initially nonintubated groups (56.5% vs. 33.3%, P = 0.110).” [page 1, line 31]
Introduction
Point 4: Line 46 – consider updating this to reflect the effectiveness of dexamethasone in critically ill patients with COVID.
Response 4: Thank you for your suggestion. We have revised the corresponding contents of the Introduction section by citing a pertinent reference as follows:
“Although it was reported that dexamethasone reduced the mortality in COVID-19 patients receiving invasive mechanical ventilation (MV) [8], due to lack of a proven effective antiviral agent to date, timely application of MV and lung protective strategies also play important role as a life-saving intervention in critically ill patients with COVID-19.” [page 2, lines 51-54]
Point 5: Line 52 – you cannot quote the Kangelaris study as reporting that “delayed intubation had markedly higher mortality rates.” Never does that study suggest that the late intubation group was synonymous with delayed intubation. Delayed intubation suggests that intubation should have been performed early but was not.
Response 5: Thank you for your insightful comment. We have corrected the word follows:
“A previous study reported that ARDS patients undergoing late intubation had markedly higher mortality rates compared to those who were intubated early in the course of the illness.” [page 2, line 59]
Materials and Methods
Point 6: Line 99 – was “same day” within a 24 hour period, starting at time of study inclusion? Or was this within 24 hours of hospital admission, ICU admission or other?
Response 6: Thank you for your valuable comment. “Same day” meant “within a 24hr period” at time of meeting ARDS criteria. We have modified the expression as follows:
“(1) Early intubation: Intubated/mechanically ventilated and meeting ARDS criteria on the same day (within 24 hours)” [page 3, lines 106-107]
Table 1
Point 7: I am confused about some of the reported p-values in this table. Take the diabetes and chronic kidney disease rows, for example. For diabetes, there are 10 patients in the early group and 8 in the initially nonintubated group with a p-value of 0.47. For kidney disease, there are 10 patients in the early group and only 2 in the initially nonintubated group which makes for a greater difference, but a much larger p-value. Am I misinterpreting the data here? Should the p-value be smaller for this 43.5% vs 8.3% difference?
Response 7: We apologize for this mistake. This error happened during the process of organizing the contents of the table before submission. We meticulously rechecked all the data and revised throughout the entire manuscript.
At this point, we think that two columns for early intubation group seem to be redundant. Thus, we have revised all the tables with repetitive early intubation group column omitted.
Results
Point 8: Line 179 Clinical outcomes – can you report the time to intubation in the late intubation group?
Response 8: Thank you for your valuable comment. We have added this data to the Results section as follows:
“The median time interval from ARDS diagnosis to intubation in the late intubation group was 3 days (IQR, 1-7 days).” [page 3, lines 132-133]
Point 9: Figure 1 – can you explain what point you want to get across with this Figure?
Response 9: We intended to visualize the main results from another angle. Never-intubated patients accounted for about 30% of those who survived from COVID-19 related ARDS. We think this figure may be omitted.
Point 10: Line 193-194 – be clear about which groups you are referring to in the parentheses (early vs initially nonintubated and early vs late intubation).
Response 10: Thank you for your suggestion. We have revised the sentence more clearly as follows:
“There was no statistically significant difference in in-hospital mortality between the early intubation and initially nonintubated groups (56.5% vs. 33.3%, P = 0.110) and between the early and late intubation groups (56.5% vs. 43.8%, P = 0.433)” [page 8, lines 216-218]
Point 11: Line 198 – should read “in” terms of
Response 11: Thank you for helpful comment. We have corrected the sentence. [page 8, line 222]
Point 12: Figure 2 – in black and white, the legend is uninterpretable.
Response 12: Thank you for your suggestion. We have modified the Figure 2 according to your comment.
Point 13: Line 216 – Because the higher mortality in the early group is fully explained by their higher severity of illness at presentation, I am not sure of the value of this analysis.
Response 13: We agree with your concern. It is clear that the severity of illness was higher in the early intubation group, and is believed to have contributed significantly to the difference in mortality. However, in our opinion, not only the severity of illness at ICU admission but also various factors related to treatment are involved in the mortality rate of critically ill patients with COVID-19, and the timing of intubation may also be one of those factors. As it is difficult for us to conduct a prospective study on this subject right away, we tried to derive the association between the timing of intubation and mortality after adjusting the variables including severity as much as possible through multivariate analysis. Of course, due to the small number of samples, this statistical adjustment is considered to be limited, and this is mentioned in the Discussion section as follows:
“Patients in the early intubation group had higher disease severity, which was estimated to have had a significant effect on the relatively higher mortality rate of those patients, and the difference in mortality rates seems not to be statistically significant due to the small sample size. Although we attempted to adjust variables including severity of illness by multivariate analysis, there may be a statistical limitation related to sample size, which should be taken into account when interpreting our findings.” [page 13, lines 323-332]
Discussion
Point 14: Line 228 – This retrospective study design cannot determine whether timing of intubation “affects” survival.
Response 14: Thank you for helpful comment. In accordance with your suggestion, we have corrected the sentence as follows:
“Our study was conducted to examine how clinical features and outcomes differed depending on the timing of intubation and to verify whether early intubation is associated with the survival of critically ill COVID-19 patients with ARDS” [page 11, line 257]
Point 15: Line 265-267 – I think that this statement goes too far and is not supported by data.
Response 15: We agree with your opinion. We have deleted the sentences as per your opinion.
Point 16: Line 275-276 – In the initially nonintubated group, if both the never intubated and the late intubated groups were both on HFNC, can you say that HFNC “may help avoid unnecessary intubation?”
Response 16: Thank you for helpful comment. We have removed the passage that could be confusing.
Point 17: 297-299 – You repeatedly state that “no significant difference in mortality between the early intubation and the other groups was observed.” However, to me there was a clear trend towards higher mortality in the early intubation group than the late intubation group (56 vs 33%) that was driven by higher disease severity and only non-significant because of small sample size. This should probably be mentioned.
Response 17: We agree with your opinion. We have removed the controversial expression and further described the limitations as stated in Response 12.
“Patients in the early intubation group had higher disease severity, which was estimated to have had a significant effect on the relatively higher mortality rate of those patients, and the difference in mortality rates seems not to be statistically significant due to the small sample size. Although we attempted to adjust variables including severity of illness by multivariate analysis, there may be a statistical limitation related to sample size, which should be taken into account when interpreting our findings.” [page 13, lines 323-332]

Reviewer 3 Report
I believe that the article is well written. It's original and very interesting. Obviously it should be better if it was a RCT and not a retrospective but it was impossible at the moment.
Author Response
Thank you for your valuable comments.
Reviewer 4 Report
- Overall a reasonable approach, conclusions and discussions for an observational design, but the title appears to completely misguide the reader (i.e this is an observational study and not one where the authors calculated power for determining the effect of early intubation on mortality). Strongly suggest to rephrase the title that explicitly implies the observational design.
- Abstract: Conclusion states in line 31 that "....early intubation had no survival benefit". This should be replaced as "not associated with improved survival "
- Materials and methods:
- Authors make no comments about their protocols and criteria for each of the following, which will benefit the readers
- Admission
- HFNC
- Intubation
- BiPAP/NIV
- How did they choose between these different options for oxygenation.
- Authors make no comments about their protocols and criteria for each of the following, which will benefit the readers
- Results:
- Were there any significant results when comparing those who were never intubated vs any intubation ?
- There is a total lack of ventilator specific data, which impacts mortality significantly in patients with ARDS. This may a have a significant negative impact on this entire study.
- Discussion is well written with very well written limitations section.
Author Response
We thank you for your constructive comments, which have helped us to improve our manuscript.
Point 1: Overall a reasonable approach, conclusions and discussions for an observational design, but the title appears to completely misguide the reader (i.e this is an observational study and not one where the authors calculated power for determining the effect of early intubation on mortality). Strongly suggest to rephrase the title that explicitly implies the observational design.
Response 1: Thank you for meaningful comment. We have changed the title of the manuscript as follows according to your suggestion: “Clinical significance of timing of intubation in critically ill patients with COVID-19: A multi-center retrospective study”
Point 2: Abstract: Conclusion states in line 31 that "....early intubation had no survival benefit". This should be replaced as "not associated with improved survival "
Response 2: Thank you for comment. As you pointed out, we have revised the sentence as follows:
“In conclusion, in this study of critically ill COVID-19 patients with ARDS, early intubation was not associated with improved survival.” [page 1, line 35-36]
The same wording is found at the end of the manuscript, so we have also modified it:
Furthermore, after adjustment for possible confounding factors, early intubation was not associated with improved survival. [page 13, line 340]
Point 3-1: Materials and methods: Authors make no comments about their protocols and criteria for each of the following, which will benefit the readers
1.Admission
2.HFNC
3.Intubation
4.BiPAP/NIV
Response 3-1: Thank you for your important comment. However, this is a retrospective study. Those matters were decided at the discretion of the attending physician without a standardized protocol. We have added a related sentence to the Materials and Methods section as follows:
“The decision of ICU admission, oxygen therapy, respiratory support, and intubation was at the discretion of the attending physician.” [page 2, lines 87-88]
Point 3-2: How did they choose between these different options for oxygenation.
Response 3-2: Likewise, it was chosen at the discretion of the attending physician. We think it is one of the limitations of retrospective research.
Point 4-1: Results: Were there any significant results when comparing those who were never intubated vs any intubation?
Response 4-1: Thank you for your comment. We have compared the clinical characteristics between the never intubated group and any intubation group as per your suggestion. APACHE II score was significantly lower (median, 10 vs. 14) and PF ratio at the time of diagnosis of ARDS was significantly higher (median, 173 vs. 95) in the never intubated group than in the any intubation group. Otherwise no significant difference was found between the groups. It was somewhat distanced from the main purpose of our study, so we did not mention in the manuscript.
Point 4-2: There is a total lack of ventilator specific data, which impacts mortality significantly in patients with ARDS. This may a have a significant negative impact on this entire study.
Response 4-2: We have added ventilator specific data to the Table 3 following your suggestion. Among them, the plateau pressure showed a significant difference between the groups, and we added this variable to the multivariate analysis, in which the result did not change significantly. We have additionally described ventilator specific data in the Results section as follows:
“Among the initial ventilator parameters, plateau pressure of the early intubation group was significantly lower than that of the late intubation group (median, 27 mmHg; IQR, 22–29 vs. 29 mmHg; IQR, 26–32; P = 0.014).” [page 7, lines 192-194]
“Variables included in the model were respiratory rate, arterial pH, PaCO2, PaO2/FiO2, use of HFNC, plateau pressure, VAP during ICU stay, and APACHE II score” [page 11, line 247]
“The same analysis was conducted on patients treated with MV and adjusted based on variables that differed between the early and late intubation groups; respiratory rate, PaCO2, use of HFNC, plateau pressure, and APACHE II score.” [page 11, line 252]
Point 5: Discussion is well written with very well written limitations section.
Response 5: Thank you.

Round 2
Reviewer 2 Report
Well done revision which included all suggested changes.
Author Response
Thank you very much indeed.
Reviewer 4 Report
The authors have amended their manuscript to a certain degree and have responded to the other comments well. However, there is a problem with their ventilator data. They have elected to include only the ventilator data at admission, which is not useful at all (everyone's FiO2 will be 100% soon after intubation) and then performed statistical tests only on these initial values, which is not helpful at all and is totally misleading. Some examples of how to display the ventilator data are available at the following links. Serial values and comparisons are important to discern their approach and infer meaningful results.
https://www.nejm.org/doi/full/10.1056/nejmoa2004500
Without providing data on serial ventilatory parameters and comparison between these two groups (mean or median), authors' statement of about no difference in outcomes becomes nebulous. What if their ventilatory strategies were not in par with recommended settings for ARDS (given that there is low adoption of lung protective strategy ventilation even in developed countries) and that was responsible for the indifference in outcomes ?
If these serial ventilator data are unavailable, then this has to be explicitly stated in their methods and limitations and consider removing the initial ventilator data, which alone does not guide the reader in any way.
